# A SUMO-dependent feedback loop senses and controls the biogenesis of nuclear pore subunits

Jérôme O. Rouvière [1,5], Manuel Bulfoni [2], Alex Tuck[3,6], Bertrand Cosson [2], Frédéric Devaux[4] & Benoit Palancade [1]

While the activity of multiprotein complexes is crucial for cellular metabolism, little is known about the mechanisms that collectively control the expression of their components. Here, we investigate the regulations targeting the biogenesis of the nuclear pore complex (NPC), the macromolecular assembly mediating nucleocytoplasmic exchanges. Systematic analysis of RNA-binding proteins interactomes, together with in vivo and in vitro assays, reveal that a subset of NPC mRNAs are specifically bound by Hek2, a yeast hnRNP K-like protein. Hek2-dependent translational repression and protein turnover are further shown to finely tune the levels of NPC subunits. Strikingly, mutations or physiological perturbations altering pore integrity decrease the levels of the NPC-associated SUMO protease Ulp1, and trigger the accumulation of sumoylated versions of Hek2 unable to bind NPC mRNAs. Our results support the existence of a quality control mechanism involving Ulp1 as a sensor of NPC integrity and Hek2 as a repressor of NPC biogenesis.

[1] Institut Jacques Monod, CNRS, UMR 7592, Univ Paris Diderot, Sorbonne Paris Cité, 15 rue Hélène Brion, 75013 Paris, France. [2] Université Paris Diderot, Sorbonne Paris Cité, Epigenetics and Cell Fate, UMR7216, CNRS, 35 rue Hélène Brion, 75013 Paris, France. [3] Wellcome Trust Centre for Cell Biology, University of Edinburgh, Max Born Crescent, Edinburgh EH9 3BF, UK. [4] Sorbonne Université, CNRS, Institut de biologie Paris-Seine (IBPS), UMR 7238, Laboratoire de biologie computationnelle et quantitative, LCQB, 4 place Jussieu, 75005 Paris, France. [5] Present address: Department of Molecular Biology and Genetics, Aarhus University, C.F. Møllers Allé 3, DK-8000 Aarhus C, Denmark. [6] Present address: Friedrich Miescher Institute for Biomedical Research, Maulbeerstrasse 66, 4058 Basel, Switzerland. Correspondence and requests for materials should be addressed to B.P. (email: benoit.palancade@ijm.fr)

Virtually all cellular processes rely on the function of multiprotein assemblies. While their stoichiometry has to be tightly controlled to prevent an imbalance of subunits that could interfere with their assembly or titrate their targets, their global abundance has also to be adjusted in response to the cellular demand[1]. Multiple layers of mechanisms have been reported to partake in the accurate biogenesis of multisubunit complexes. First, all the steps in the gene expression pathway, including messenger RNA (mRNA) synthesis, processing, transport, stability and translation, can be regulated in a coordinate manner, either to lead to the proportional synthesis of the different subunits of multiprotein assemblies, a prominent strategy in prokaryotes[2], or to respond to environmental or physiological cues, as exemplified by the ribosome biosynthesis pathway[3]. In this frame, a pivotal role has emerged for transcriptional regulators and RNA-binding proteins, the latter being in particular capable to tune the translation rate of their target messenger ribonucleoparticles (mRNPs). Second, molecular chaperones and assembly factors can further assist the assembly of multiprotein complexes, as also described for ribosomes[3], in some cases in a cotranslational manner[4]. Finally, excess complexes or unassembled, orphan polypeptides can be targeted for degradation by the proteasome or the lysosome[5], with these quality control processes being critical to adjust stoichiometry and to cope with altered protein dosage[6,7]. However, despite our improved knowledge in proteome dynamics, the specific mechanisms at play for most multiprotein complexes remain largely unknown.

The nuclear pore complex (NPC) provides a paradigmatic example of an essential multisubunit complex whose homeostasis is crucial yet poorly understood. NPCs are megadalton-sized

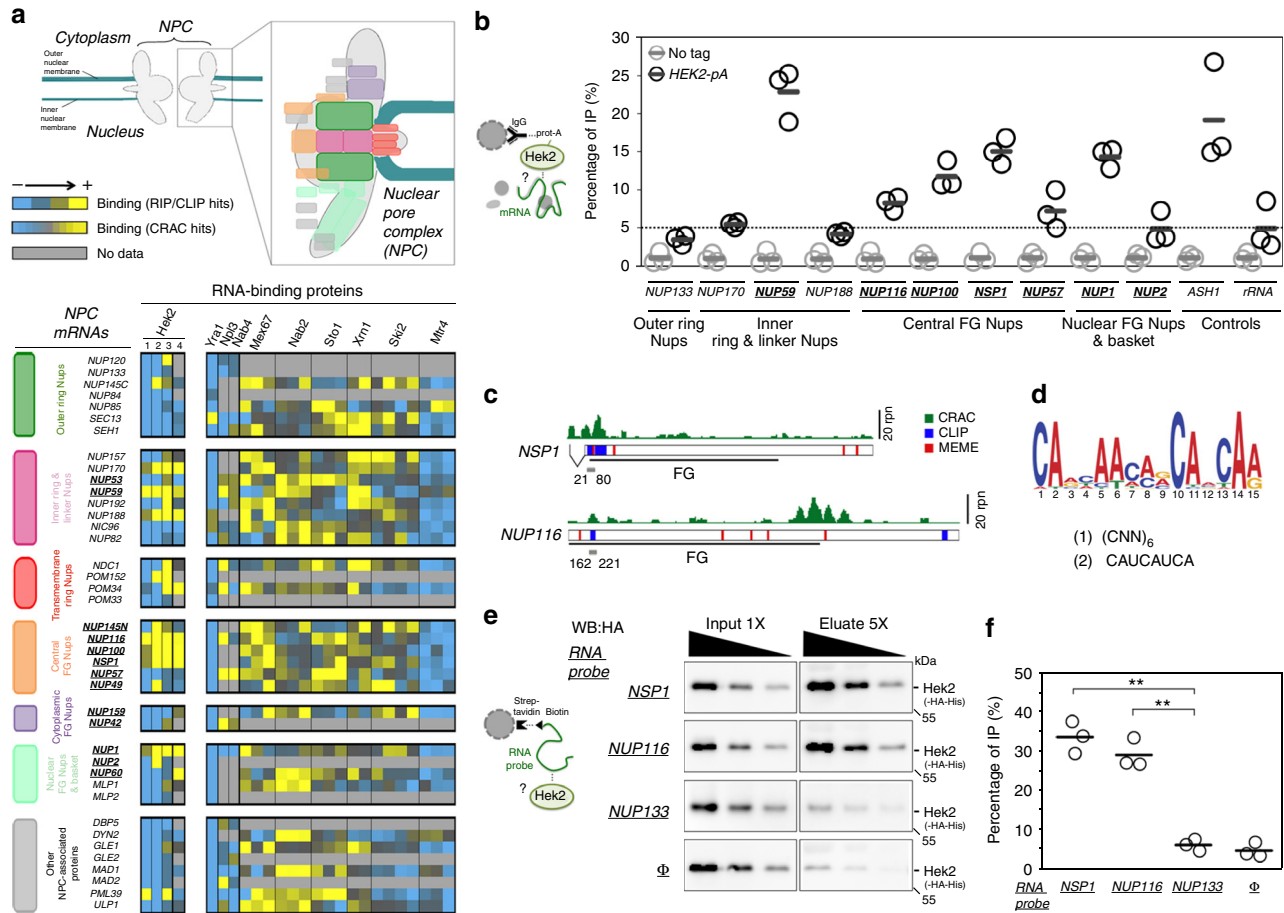

**Fig. 1** The hnRNP K-like protein Hek2 specifically associates with a subset of *NPC* mRNAs. **a** Top, Representation of the yeast nuclear pore complex (NPC), showing subcomplexes as colored boxes. Bottom, mRNAs encoding NPC components are sorted by subcomplexes and the strength of their association to the different indicated RNA-binding proteins (RBP) is represented by a color code, as scored in distinct RIP, CLIP or CRAC datasets. Bright yellow indicates the preferred association of a given mRNA to the RBP of interest. For Sto1, Mtr4, Nab2, Mex67, Xrn1 and Ski2, multiple repetitions are displayed[27]. For Hek2, the results from independent studies are represented: (1)[24], (2)[26], (3)[25], (4)[27]. FG-Nups appear in bold, underlined. The *NUP145* mRNA gives rise to both Nup145-N and Nup145-C nucleoporins and is displayed for each of the according subcomplexes. **b** Hek2-pA-associated mRNAs were immunopurified and quantified by RT-qPCR using specific primer pairs. Percentages of IP are the ratios between purified and input RNAs, normalized to the amount of purified bait and set to 1 for the "no tag". Means and individual points (n = 3) are displayed. A schematic representation of the assay is shown. **c** Overview of Hek2-binding sites on *NSP1* and *NUP116* mRNAs. The number of CRAC hits (rpn (reads per nucleotide))[27], the position of CLIP fragments[26] and the occurrences of the binding site found by the MEME analysis are indicated. The positions of the FG-coding region and of minimal Hek2-binding sites used for in vitro pull down (in gray) are represented. The broken line indicates the *NSP1* intron. **d** MEME result from *NUP59*, *NUP116*, *NUP1*, *NSP1* and *NUP100* sequences. (1, 2): previously identified Hek2-binding sequences[24,26]. **e** Left, Schematic representation of the assay. Recombinant HA-tagged Hek2 was incubated with streptavidin beads either naive (Φ) or coated with biotinylated RNA probes encompassing Hek2-binding sites from *NSP1* (21–80) or *NUP116* (162–221) or a sequence from *NUP133* (1429–1488). Right, Decreasing amounts of input and eluate fractions were loaded for quantification. **f** Percentages of IP are the ratios between Hek2 amounts in the eluate and input fractions, calculated from (**e**). Means and individual points (n = 3) are displayed. **P < 0.01 (Welch's *t*-test)

proteinaceous assemblies embedded at the fusion points of the nuclear envelope and formed of modular repeats of ~30 distinct protein subunits—the nucleoporins (Nups)—which assemble within subcomplexes and organize with a 8-fold rotational symmetry[8]. The major task of NPCs is the selective nucleocytoplasmic transport of macromolecules, i.e., proteins and RNA-containing particles, a process involving dynamic interactions between the cargo-transport factor complexes and the phenylalanine-glycine (FG) repeats-harboring nucleoporins that lie within the central channel and the peripheral extensions of the NPC[9]. The stepwise assembly of nucleoporins to build complete NPCs proceeds through defined pathways, either following mitosis in conjunction with nuclear envelope reformation or during interphase, the unique assembly mode compatible with the closed mitosis of fungi. Nucleoporins themselves are essential players in NPC assembly, either through scaffolding or by mediating interactions with chromatin and/or membranes. In addition, non-NPC factors, such as membrane bending proteins, also contribute to NPC biogenesis[10]. While multiple studies have depicted the choreography of NPC assembly, together with their structural organization, little is known about the mechanisms that sustain the timely production of stoichiometric amounts of Nups or that could possibly sense and adjust NPC biogenesis depending on cell physiology.

The high connectivity observed between NPCs and several biological processes could place them in a strategical position to communicate their status to the cell. Indeed, NPCs have been described to contribute to multiple aspects of transcriptional regulation, genome stability and cell cycle progression[9]. In some situations, these connections are mediated by physical interactions between NPCs and enzymes of the small ubiquitin-related modifier (SUMO) pathway[11]. Sumoylation is a post-translational modification that can modulate the binding properties or the conformation of its targets, ultimately impacting their stability, their localization or their biological activity[12]. Among the distinct enzymes of the sumoylation/desumoylation machinery shown to associate with NPCs, the conserved SUMO protease Ulp1 has essential functions in SUMO processing and deconjugation in budding yeast. The docking of this enzyme to the nucleoplasmic side of NPCs is essential for viability[13,14] and is believed to involve its nuclear import through karyopherins, followed by its association with several nucleoporins[15–19]. Proper NPC localization of Ulp1 has been shown to be critical for the spatio-temporal control of the sumoylation of certain targets, some of them being important for genetic integrity or gene regulation[13,16,20,21].

Here, we report an original mechanism by which the synthesis of NPC subunits is regulated in response to changes in NPC integrity in budding yeast. We show that a subset of Nup-encoding mRNAs is defined by the specific binding of the translational regulator Hek2. Hek2-regulated NPC mRNA translation and protein turnover are further shown to finely tune the levels of the corresponding nucleoporins. Strikingly, Hek2 binding to NPC mRNAs is prevented by sumoylation, a process reversed by the SUMO protease Ulp1. Mutant or physiological situations in which NPC functionality is compromised are associated with the loss of Ulp1 activity and the subsequent accumulation of sumoylated Hek2 versions that are inactive for NPC mRNA translational repression. We propose that Ulp1 and Hek2 are respectively the sensor and the effector of a feedback loop maintaining nucleoporin homeostasis.

## Results

**A unique mRNP composition for a subset of *NPC* mRNAs.** In order to unravel novel mechanisms regulating NPC biogenesis, we systematically analyzed the association of Nup-encoding

(*NPC*) mRNAs with different RNA-binding proteins (RBPs) in budding yeast. For this purpose, we took advantage of previously published large-scale datasets obtained through RNA immuno-precipitation (RIP)[22–25], crosslinking immunoprecipitation (CLIP)[26] or crosslinking and analysis of complementary DNA (CRAC)[27]. We collected the association data for 39 *NPC* mRNAs (encoding Nups and NPC-associated proteins, Fig. 1a and Supplementary Fig. 1a) with a panel of 10 mRNA-associated factors involved in different stages of mRNA metabolism, including assembly into mRNP (Sto1), processing (Npl3, Nab4/Hrp1), nuclear export (Yra1, Nab2, Mex67), degradation (Xrn1, Ski2, Mtr4) or mRNA localization/translation (Hek2) (Fig. 1a). This analysis revealed that *NPC* mRNAs have generally the same typical features of expressed, protein-coding RNAs, e.g., they readily associate with mRNA export factors (Mex67, Nab2), but not with the non-coding RNA degradation machinery (Mtr4) (Fig. 1a, bottom right panel). Strikingly however, a small subset of *NPC* mRNAs (namely *NUP170*, *NUP59*, *NUP188*, *NUP116*, *NUP100*, *NSP1* and *NUP1*) appeared to specifically bind the conserved *He*terogeneous nuclear ribonucleoprotein *K*-like factor Hek2 (a.k.a. Khd1[28,29]), a feature detected in four independent datasets (Fig. 1a, bottom left panel). The enrichment of certain *NPC* mRNAs among Hek2-bound targets appeared significant in a Gene Set Enrichment Analysis ($P = 0.02$) and was neither a mere consequence of the different expression levels of these particular transcripts (Supplementary Fig. 1b) nor a general feature of any multiprotein complexes, since it was not observed when similar analyses were performed for mRNAs encoding proteasome or exosome subunits (Supplementary Fig. 1c).

To further validate this finding in vivo, we immunoprecipitated a protein A-tagged version of Hek2 from yeast cells and analyzed its interaction with *NPC* mRNAs by reverse transcription-quantitative polymerase chain reaction (RT-qPCR). In agreement with our previous findings, Hek2 preferentially associated with *NUP59*, *NUP116*, *NUP100*, *NSP1* and *NUP1* mRNAs (Fig. 1b), to a similar extent as its prototypal target *ASH1*[28,29], but not with *NUP133*, *NUP57* or *NUP2* mRNAs (Fig. 1b), for which Hek2 binding was in the same range as its reported, unclear association to rRNA[27]. Preferential binding to *NUP170* and *NUP188* mRNAs was not confirmed, with the previous finding from genome-wide studies possibly reflecting their different expression levels in other genetic backgrounds. In contrast, immunoprecipitation of Hpr1, a subunit of the mRNP packaging THO complex, did not reveal any similar preferred association to a subset of *NPC* mRNAs (Supplementary Fig. 1d).

We then asked whether Hek2 was directly associating to this subset of *NPC* mRNAs (i.e., *NUP59*, *NUP116*, *NUP100*, *NSP1* and *NUP1*), as expected from CLIP/CRAC studies[26,27]. To this aim, we first delineated Hek2-binding sites on these mRNAs by mining CLIP/CRAC data (Fig. 1c) and by searching their sequences for common motifs using the MEME software (Fig. 1c, d). This in silico approach revealed that these mRNAs share a common CA-rich motif (Fig. 1d), similar to the two previously reported Hek2-binding sites, i.e. $(CNN)_6$[24] and CAUCAUCA[26]. As anticipated from a previous study[26], this motif was overlapping some but not all in vivo Hek2-binding peaks as defined by CLIP or CRAC, allowing us to define putative minimal bound domains in *NSP1* and *NUP116* mRNAs (Fig. 1c, gray bars). In an in vitro binding assay, synthetic biotinylated RNA probes encompassing these Hek2-binding sequences were further found to specifically pull down recombinant, purified Hek2 (Fig. 1e, f), but not a control protein (Supplementary Fig. 1e).

Altogether, our data establish that a direct association with the hnRNP Hek2 specifically defines a subset of *NPC* mRNPs. Notably, the five Hek2-bound *NPC* mRNAs are coding for FG-Nups, which are critical for nucleocytoplasmic transport[30].

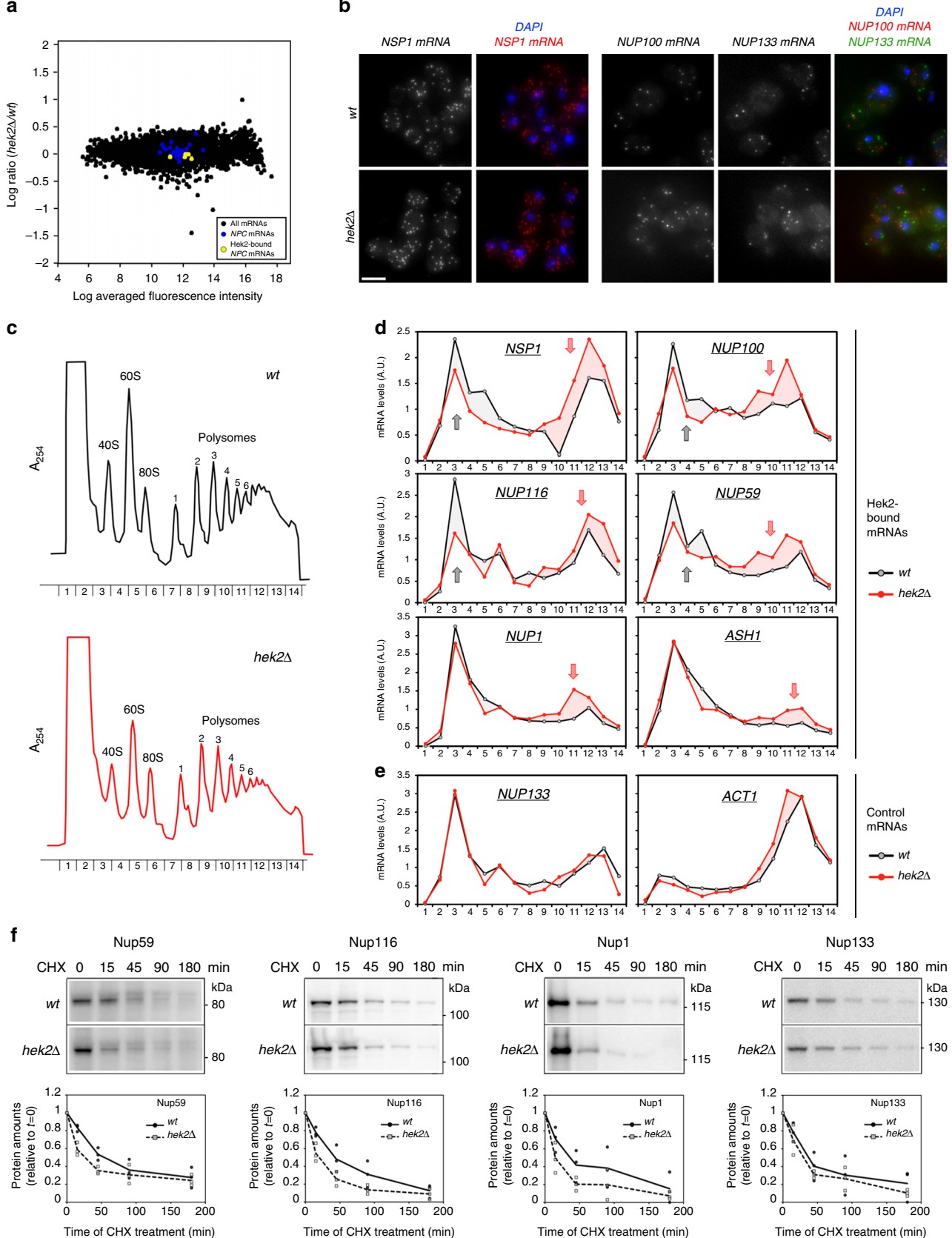

**Fig. 2** Hek2-dependent translational repression and protein turnover define nucleoporin levels. **a** Transcriptome analysis of the *hek2Δ* mutant. The *y*-axis is the averaged log2 of the *hek2Δ/wt* ratios calculated from two independent microarray hybridizations. The *x*-axis is the log2 of the averaged fluorescence intensities. mRNAs encoding NPCs components are colored depending on their association to Hek2 (from Fig. 1). **b** Single-molecule FISH was performed on *wt* and *hek2Δ* cells using set of probes specific for the indicated mRNAs. *NSP1* and *NUP100* probes were coupled to the Quasar570 fluorophore (red), and *NUP133* probes to Quasar670 (far red). The *z*-projections are displayed, together with merged images with a nuclear staining (DAPI). Scale bar, 5 μm. **c** Polysome fractionation from *wt* and *hek2Δ* cells (W303 background). The absorbance at 254 nm ($A_{254}$) recorded during the collection of the fractions of the gradient is displayed. The positions of 40S, 60S, 80S ribosomal species are indicated, as well as the number of ribosomes per mRNA in polysomes fractions. **d** Relative distribution of the indicated mRNAs in polysome gradients from *wt* (black lines) and *hek2Δ* (red lines) cells. mRNAs amounts in each fraction were quantified by RT-qPCR, normalized to the sum of the fractions and to the distribution of a control spike RNA. Gray arrows indicate a decrease in the amounts of mRNAs found in the light fractions in *hek2Δ* cells, while red arrows point to an increase in the quantity of mRNAs found in the polysomes fractions. These results are representative of four independent experiments (two performed in the W303 background, two in the BY4742 background; see Supplementary Fig. 2). **e** Same as (**d**) for *NUP133* and *ACT1* control mRNAs. **f** Protein levels of the indicated nucleoporins (Nup116, Nup1, Nup133) and of a GFP-tagged version of Nup59 were scored in *wt* and *hek2Δ* cells treated with cycloheximide (CHX) for the indicated time (min). Top, Whole-cell extracts were analyzed by western blotting using anti-GFP, anti-GLFG, anti-FSFG or anti-Nup133 antibodies. Bottom, The relative amounts of the indicated proteins (mean and individual points; $n = 3$) were quantified over the time following CHX treatment and expressed relative to $t = 0$

**A role for Hek2 in the metabolism of *NPC* mRNAs**. We further investigated how Hek2 binding impacts the fate of these particular *NPC* mRNAs. While previous studies have revealed that Hek2 associates with an important fraction of the transcriptome, the consequences of this recruitment for mRNA metabolism have only been documented in a few situations where Hek2 binding can cause increased mRNA stability[24], asymmetrical localization[28] or translational repression[26,29].

To determine whether Hek2 binding influences the steady-state levels of *NPC* mRNAs, we first profiled the transcriptome of *hek2Δ* mutant yeast cells (Fig. 2a). Genome-wide, Hek2-bound mRNAs showed a tendency to be less abundant upon Hek2 inactivation (Supplementary Fig. 2a), a trend not observed for Nab2-associated transcripts (Supplementary Fig. 2b), highlighting the sensitivity and the specificity of our analysis. However, *NPC* mRNAs levels were not significantly affected by the absence of Hek2, whether or not they associate with this factor (Fig. 2a). We then compared the localization of *NPC* mRNAs in *wt* and *hek2Δ* cells using single-molecule fluorescence in situ hybridization (smFISH; Fig. 2b). Detection of *NSP1*, *NUP100* and *NUP133* mRNAs using specific sets of probes revealed a punctuate, cytoplasmic localization for these Nup-encoding transcripts in *wt* cells (Fig. 2b, top panels). Upon *HEK2* deletion, this random distribution, as well as the total number of detected RNA dots, were unchanged for both Hek2-bound (*NSP1*, *NUP100*) and Hek2-unbound (*NUP133*) mRNAs (Fig. 2b, bottom panels). This set of data therefore establishes that Hek2 binding modulates neither the levels nor the localization of *NPC* mRNAs.

We then monitored the possible influence of Hek2 on *NPC* mRNA translation using polysome fractionation on sucrose gradients, which resolve free mRNPs and ribosomal subunits from translation-engaged mRNAs (Fig. 2c, Supplementary Fig. 2c). RT-qPCR analysis of the fractions of the *wt* polysome gradient revealed a bimodal distribution for Hek2-bound (Fig. 2d, Supplementary Fig. 2d, black lines) and Hek2–unbound (Fig. 2e, Supplementary Fig. 2e, black lines) *NPC* mRNAs. The largest fraction of *NPC* mRNAs migrated in the lightest fractions (*#1–6*), corresponding to free, untranslated mRNPs and resembling the pattern observed for the repressed *ASH1* mRNA (Fig. 2d). A less abundant fraction of *NPC* mRNAs peaked with polysome-containing fractions (*#9–13*), similar to the peak of the well-translated *ACT1* mRNA (Fig. 2e). Further analysis of the polysome profile from *hek2Δ* cells did not reveal any differences in the distribution of ribosomal species as compared to *wt* cells (Fig. 2c, Supplementary Fig. 2c). Strikingly, *HEK2* inactivation decreased the amounts of translationally repressed Hek2-bound *NPC* mRNAs (Fig. 2d, gray arrows) and triggered their

redistribution in the translated population, with a peak in heavy polysomes fractions (≥4 ribosomes/mRNA; Fig. 2d, red arrows). This behavior was similar to the one reported for the Hek2-repressed *ASH1* mRNA[29] (see also Fig. 2d) and was not observed for mRNAs which are not bound by Hek2 (e.g., *NUP133* and *ACT1*, Fig. 2e).

Having established that Hek2 binding onto its *NPC* target mRNAs contributes to their maintenance in a translationally repressed state, we wondered whether it would affect the raw levels of their cognate nucleoporins. Notably, *HEK2* inactivation, while increasing the fraction of translated *NUP59*, *NUP116*, or *NUP1* mRNAs (Fig. 2d), did not trigger any drastic changes in the steady-state levels of the corresponding nucleoporins (see $t = 0$ in Fig. 2f). Since excess synthesis of subunits of multiprotein complexes can be buffered by increased protein turnover[6], we monitored the half-lives of these nucleoporins in *wt* and *hek2Δ* cells. Strikingly, the degradation rates of the three nucleoporins, as estimated from cycloheximide chase experiments, were higher in the absence of Hek2 (Fig. 2f), revealing that the enhanced synthesis of nucleoporins is attenuated by their increased turnover in these mutant cells. Consistently, the kinetics of degradation of Nup133, whose translation is independent from Hek2 activity, was unaffected in *hek2Δ* cells (Fig. 2f). The raw levels of this subset of nucleoporins are thereby tightly controlled by both Hek2-mediated translational control and protein degradation.

The latter results suggested that Hek2 function might become crucial in conditions of disturbed proteostasis. To test this hypothesis, we combined *HEK2* inactivation with MG132-mediated inhibition of proteasomal degradation in drug-sensitive yeast strains, and further analyzed the cellular localization of Nup1, whose overexpression was previously reported to give rise to lethality[31]. Strikingly, simultaneous inhibition of Hek2 and proteasome functions enhanced the formation of abnormal cytoplasmic foci of this nucleoporin in a small fraction of cells (Supplementary Fig. 2f). The fine-tuning of nucleoporin amounts mediated by Hek2 translational repression and proteasome-dependent turnover can thereby be critical to prevent the accumulation of mislocalized NPC subunits.

**Hek2 can be modified by SUMO**. Having established that Hek2 can prevent excess Nup production, we then wondered whether regulatory mechanisms could reverse this repressing activity in response to an increased cellular demand for nucleoporins. Yck1-mediated phosphorylation of Hek2 was previously reported to disrupt its association with the *ASH1* mRNA at the bud cortex where this asymmetrically localized mRNA is targeted[29]. However, this plasma membrane-anchored kinase is unlikely to

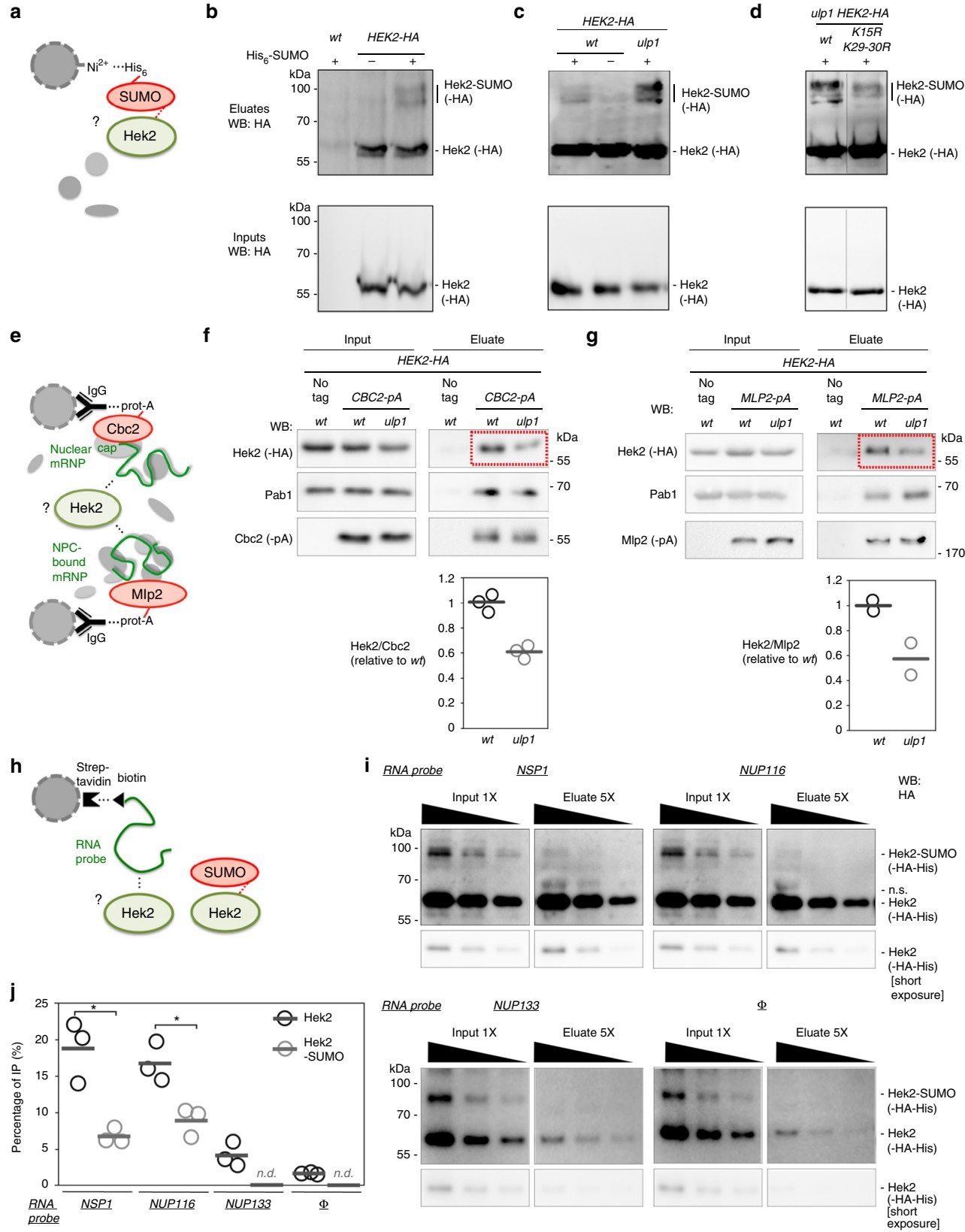

similarly target cytoplasm-localized *NPC* mRNPs (Fig. 2b). In view of the functional relationships between sumoylation and NPCs[11] and of the multiple examples of nucleic acid-binding proteins whose activity is controlled by SUMO[32], we rather wondered whether Hek2 could be regulated by this modification.

To answer this question, cellular SUMO conjugates were purified by denaturing $Ni^{2+}$ chromatography from strains expressing a poly-histidine-tagged version of SUMO and the hemagglutinin (HA)-tagged version of Hek2 (Fig. 3a). This assay specifically detected slower-migrating species of Hek2 in the

**Fig. 3** Hek2 sumoylation prevents its association to mRNAs. **a** Principle of the purification of sumoylated Hek2. Extracts from cells expressing a His-tagged version of SUMO were used for denaturing nickel chromatography. **b**–**d** Extracts from *wt* and *HEK2-HA* cells (**b**), *HEK2-HA* and *HEK2-HA ulp1* cells (**c**) or *HEK2-HA ulp1* and *HEK2 K15R K29-30R-HA ulp1* cells (**d**) expressing or not His$_6$-SUMO ($+/-$) were used for nickel chromatography. Total lysates ("Inputs") and purified His-SUMO conjugates ("Eluates") were analyzed by western blotting using anti-HA antibodies. The positions of the sumoylated and unmodified versions of Hek2-HA, as well as molecular weights, are indicated. Note the non-specific binding of a fraction of non-sumoylated Hek2-HA (also observed in the absence of His-SUMO, second lanes in (**b**, **c**)), a classical issue in SUMO-conjugates purification. **e** Principle of the mRNP purification procedure. Cbc2 or Mlp2 are purified through a protein-A tag, and the protein content of the associated mRNPs is analyzed by western blot. Note that RNAse A treatment experiments confirmed the RNA dependence of the interactions scored in such assays[35]. **f**, **g** Top, Soluble extracts ("Input", left panels) and Cbc2-pA-associated mRNPs (**f**) or Mlp2-pA-associated mRNPs (**g**) ("Eluate", right panels) isolated from *wt* and *ulp1* cells were analyzed by immunoblotting using the indicated antibodies. Bottom, The relative amounts of Hek2 associated to Cbc2- and Mlp2-bound mRNPs are represented (mean and individual points; $n = 3$ for Cbc2-pA, $n = 2$ for Mlp2-pA). **h** Principle of the in vitro RNA-binding assay. **i** An in vitro sumoylation mixture containing both unmodified and sumoylated Hek2 was incubated with streptavidin beads either naive ($\Phi$) or previously coated with biotinylated RNA probes encompassing Hek2-binding sites from *NSP1* or *NUP116* or a sequence from *NUP133*. Decreasing amounts of input and eluate fractions were loaded to allow quantification. **j** Percentages of IP are the ratios between unmodified (or sumoylated) Hek2 amounts in the eluate and in the input fractions and were calculated from (**i**). Means and individual points ($n = 3$) are displayed. Note that sumoylated Hek2 was not detectable (n.d.) and thereby not quantified on control pull downs. *$P < 0.05$ (Welch's *t*-test)

SUMO-conjugate fraction of cells co-expressing Hek2-HA and His-SUMO (Fig. 3b). Importantly, these modified Hek2 forms were not detected upon inactivation of the unique SUMO-conjugating enzyme Ubc9 (Supplementary Fig. 3a). Conversely, these species accumulated in cells carrying a thermosensitive allele of the NPC-associated SUMO-protease Ulp1 (*ulp1-333*[33], reported to disturb both Ulp1 activity and NPC localization, and thereafter referred as *ulp1*; Fig. 3c). This pattern was not observed upon inactivation of Ulp2, the alternative yeast SUMO-deconjugating enzyme localized in the nucleoplasm[18,34] (Supplementary Fig. 3b). Furthermore, modified species accumulating in the *ulp1* mutant migrated slightly slower when they were purified from cells expressing doubly tagged His-Flag-SUMO (Supplementary Fig. 3c). Taken together, these data demonstrate the existence of SUMO-modified versions of Hek2 that are deconjugated by Ulp1 in a specific manner.

The apparent molecular weights of these Hek2 forms are compatible with mono-sumoylations occurring on distinct lysine residues. To identify their positions, we generated several plasmid-based *hek2* mutants where multiple lysines were mutated to arginines to prevent SUMO conjugation without disturbing the charge of the protein (Supplementary Fig. 3d), and expressed them in *hek2Δ* cells. While mutations of all Hek2 lysines (*K1-30R*) completely abolished sumoylation, mutations of residues 19 to 30 (*K19-30R*), 25 to 30 (*K25-30R*) or 29/30 (*K29-30R*) were found to prevent the formation of most of the lower sumoylated version of Hek2 (Supplementary Fig. 3e, lanes 5, 15, 32, 35), and mutations of lysines 8 to 18 (*K8-18R*), 13 to 18 (*K13-18R*) or 15 alone (*K15R*) strongly decreased its major upper sumoylation band (Supplementary Fig. 3e, lanes 4, 13, 22, 24). Consistently, the *K15R K29-30R* combined mutant strongly reduced Hek2 sumoylation (Fig. 3d). Importantly, the turnover of Hek2 was unaffected in conditions where its sumoylation was enhanced (*ulp1* cells) or decreased (*hek2-K15 K29-30R* cells), demonstrating that this modification does not regulate its stability (Supplementary Fig. 3f).

**Hek2 binding to *NPC* mRNAs requires desumoylation by Ulp1.** In order to determine whether Hek2 sumoylation could rather regulate its interaction with its target mRNAs, we combined the following approaches. First, we purified two different subsets of mRNPs from *wt* and *ulp1* cells and analyzed their association with Hek2 (Fig. 3e). mRNPs were isolated using as baits either Cbc2, a subunit of the nuclear cap-binding complex (Cbc2-pA, Fig. 3f), or Mlp2, which anchors mRNPs to NPCs prior to nuclear export (Mlp2-pA, Fig. 3g)[35]. Strikingly, *ULP1* loss of function triggered a clear decrease in the amounts of Hek2

recovered in both mRNP populations (Fig. 3f, g), while it did not affect the recruitment of canonical mRNP components such as the poly-A-binding protein Pab1, in agreement with our previous study[35]. Second, we specifically looked at the association of Hek2 with *NPC* mRNAs in *wt* and *ulp1* cells through Hek2-pA immunoprecipitation followed by RT-qPCR. This assay further confirmed that *ULP1* inactivation leads to a decrease in the association of Hek2 with its target mRNAs (Supplementary Fig. 3g).

These two experiments demonstrate that the SUMO protease Ulp1 is required for both Hek2 desumoylation and binding to *NPC* mRNAs, suggesting that this association could be directly repressed by SUMO. To further challenge this hypothesis, we went on to compare the binding of unmodified and sumoylated Hek2 to *NPC* mRNAs in a reconstituted in vitro assay (Fig. 3h). For this purpose, we first achieved the in vitro sumoylation of recombinant Hek2 in the presence of purified versions of the SUMO-activating enzyme (Aos1-Uba2), the SUMO-conjugating enzyme (Ubc9) and SUMO, partly reproducing the observed in vivo sumoylation pattern (Supplementary Fig. 3h, first lane). When further used in the in vitro RNA-binding assay, the sumoylated version of Hek2 was unambiguously less prone to bind RNA that its unmodified counterpart (Fig. 3i, j). Altogether, our data thereby establish that Hek2 sumoylation negatively regulates its association to *NPC* mRNAs and that Ulp1 desumoylating activity is required for optimal binding.

**Compromised NPC integrity alters Ulp1 and Hek2 activities.** The fact that the SUMO protease that controls the binding of Hek2 to *NPC* mRNAs is itself associated to nuclear pores prompted us to test whether it could be part of a feedback mechanism sensing NPC integrity and further modulating Nups biogenesis. We therefore asked whether mutant or physiological situations associated with defects in nuclear pore functions would result in changes in the activity of Ulp1 towards Hek2.

Mutants of distinct NPC subcomplexes, e.g., the outer ring Nup84 complex and the nuclear basket Nup60-Mlp1/2 complex, were previously shown to exhibit decreased levels of Ulp1 at the nuclear envelope[15,16]. To complement these findings, we systematically analyzed the localization of Ulp1 in *ΔFG* mutants in which the genetic removal of FG domains from specific nucleoporins leads to defects in nucleocytoplasmic transport, including karyopherin-dependent import[30]. In *wt* cells, the green fluorescent protein (GFP)-tagged version of Ulp1 exhibited a discontinuous rim-like staining of the nuclear periphery typical of its NPC-associated localization (Fig. 4a). In most *ΔFG* mutants however, the Ulp1-GFP nuclear envelope staining was

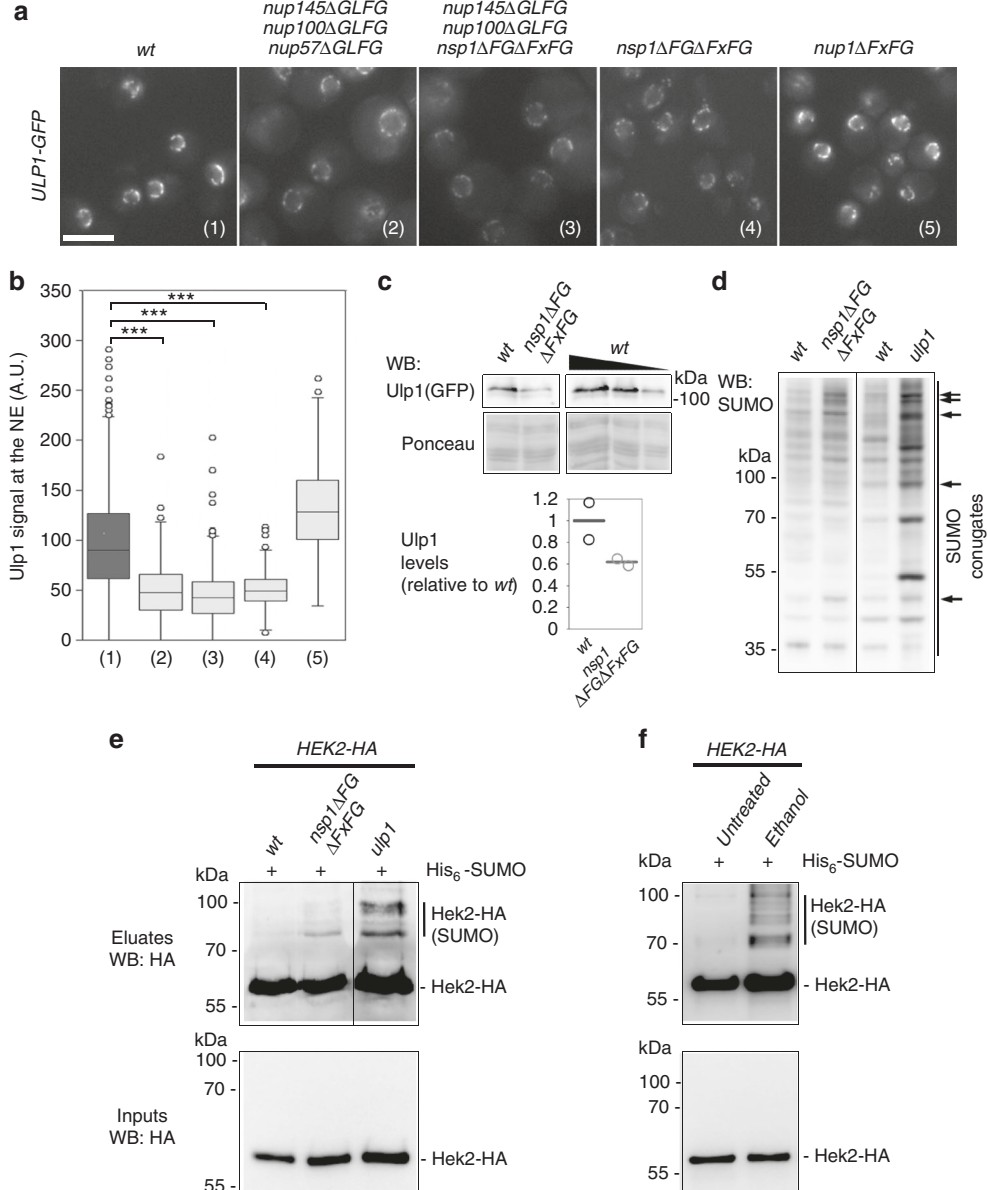

**Fig. 4** Defects in nuclear pore integrity impact Ulp1 activity and Hek2 sumoylation. **a** Fluorescence microscopy analysis of Ulp1-GFP in *wt*, *nup145ΔGLFG nup100ΔGLFG nup57ΔGLFG*, *nup145ΔGLFG nup100ΔGLFG nsp1ΔFGΔFxFG*, *nsp1ΔFG·FxFG* and *nup1ΔFxFG* cells grown at 30 °C. Scale bar, 5 μm. **b** Quantification of the Ulp1 nuclear envelope fluorescence intensity in the different strains. The numbers refer to the genotypes as depicted in (**a**). For each strain, at least 150 cells were analyzed. Boxplots were generated using KaleidaGraph (Synergy Software): each box encloses 50% of the measured values, the median is displayed as a line, and the bars extending from the top and bottom of each box mark the minimum and maximum values within the dataset falling within an acceptable range. Values falling outside of this range are displayed as individual points. ***$P < 0.001$ (Mann–Whitney–Wilcoxon test). **c** Ulp1-GFP amounts were measured in *wt* and *nsp1ΔFGΔFxFG* cells by western blotting using anti-GFP antibodies (top panel). Ponceau staining was used as a loading control (lower panel). A serial dilution of the *wt* sample was used for quantification. Ulp1-GFP amounts normalized to ponceau are represented (mean and individual points, $n = 2$). **d** Whole cell extracts of the indicated strains were analyzed by western blotting using anti-SUMO antibodies. The bands that are modified in the *nsp1ΔFGΔFxFG* mutant are also typically altered in *ulp1* cells (arrows). **e** Hek2 sumoylation was detected in *wt* and *nsp1ΔFGΔFxFG* cells as in Fig. 3. Total lysates ("Inputs") and purified His-SUMO conjugates ("Eluates") were analyzed by western blotting using anti-HA antibodies. The pattern of Hek2 sumoylation in *ulp1* cells was analyzed as a control. The positions of the sumoylated and unmodified versions of Hek2-HA, as well as molecular weights, are indicated. **f** Hek2 sumoylation was similarly detected in *wt* cells, either untreated, or treated with 10% ethanol for 1 h

significantly reduced (Fig. 4a, b). This phenotype was unlikely to be caused by a reduction in the number of NPCs, according to a previous characterization of these mutants[30], but rather reflected a decrease in the karyopherin-dependent import step that precedes Ulp1 anchoring at NPCs. Consistently, we did not observe this reduced Ulp1 staining in the *nup1ΔFG* mutant (Fig. 4a, b) which is unexpected to impair karyopherin function[30].

To further characterize this phenotype, we pursued the analysis of the *nsp1ΔFGΔFxFG* mutant in which removal of the FG domains from a single nucleoporin is sufficient to decrease Ulp1 levels at the nuclear envelope (Fig. 4a, b). In agreement with the previously reported interdependence between Ulp1 NPC localization and stability[15,16], western blot analysis of this *nsp1ΔFGΔFxFG* mutant further revealed a reduction in the total

amounts of cellular Ulp1 as compared to *wt* cells (Fig. 4c). Consistently, analysis of the global pattern of cellular SUMO conjugation in this same mutant highlighted a number of discrete changes, in particular the accumulation of high-molecular-weight SUMO conjugates, resembling those caused by *ULP1* inactivation (Fig. 4d, arrows). We then wondered whether the changes in Ulp1 levels and activity detected in this mutant were sufficient to modulate Hek2 sumoylation. Remarkably, *nsp1ΔFGΔFxFG* cells exhibited a clear increase in the levels of sumoylated Hek2 (Fig. 4e). Loss of NPC integrity upon genetic alteration of several distinct NPC components can therefore impact the levels of active Ulp1, which is sufficient to trigger the accumulation of sumoylated, inactive versions of Hek2.

We finally asked whether physiological changes in NPC integrity could also lead to the accumulation of inactive Hek2 in *wt* cells. Environmental stresses can trigger changes in NPC integrity, as exemplified by the specific delocalization of certain NPC components, including Ulp1, upon exposition to elevated alcohol levels[36–38]. We then analyzed the sumoylation levels of Hek2 in *wt* cells exposed to ethanol stress (Fig. 4f). Strikingly, increased levels of sumoylated Hek2 were detected in this situation (Fig. 4f). Changes in NPC integrity, triggered by either genetic alterations or physiological changes, can thereby translate into the accumulation of inactive versions of Hek2.

## Discussion

By combining the analysis of genomic data with in vivo and in vitro interaction assays, we have established that a subset of the mRNAs that encode the subunits of nuclear pores display a unique mRNP composition characterized by the binding of the hnRNP Hek2/Khd1 (Fig. 1). This conserved RNA-binding protein was previously reported to have various effects on the metabolism of its target mRNAs[24,26,28,29], possibly reflecting coregulations involving other RBPs[39], including the Hek2 paralog Pbp2/Hek1, or transcript specificities, as in the case of the bud-localized mRNA *ASH1*. Here, we show that Hek2 binding to Nup-encoding mRNAs affects neither their steady-state levels nor their subcellular localization (Fig. 2a, b), in contrast with other target mRNAs (Supplementary Fig. 2a-b)[28]. However, Hek2 binding appears to regulate the translation of *NPC* mRNAs. Indeed, upon *HEK2* inactivation, the percentage of translated Hek2 target mRNAs increases and peaks with the heavy polysomes containing the most actively translating ribosomes, a phenotype that is not observed for control transcripts (Fig. 2d, e). In this frame, the regulation of *NPC* mRNAs is reminiscent of the one scored for *ASH1* and *FLO11*, two mRNAs for which Hek2 binding represses translation initiation (Fig. 2d)[26,29]. In the case of the *ASH1* transcript, it was demonstrated that Hek2 directly binds to the translation factor eIF4G1, likely constraining its initiation-promoting activity[29], a mechanism of repression possibly also at play on *NPC* mRNAs. Notably, our study uncovers that in *wt* cells, these mRNAs distribute in two populations, one being actively translated and the other translationally repressed. Such a bimodal distribution is rather uncommon in yeast, in which whole-genome polysomal profiles previously revealed that most mRNAs are associated with translating ribosomes during exponential growth[40], and likely indicates undergoing translational controls. However, it has to be noted that Hek2 binding is unlikely to be the only determinant of this particular translational regulation. Indeed, a large fraction of each Hek2-bound mRNAs (e.g., *NSP1* and *NUP1*, Fig. 2d) remains untranslated in the absence of Hek2. In addition, the *NPC* mRNAs that are not among Hek2 preferred targets (e.g., *NUP133*, Fig. 2e) also exist for the most part in a translation-inactive fraction. Whether alternate RBPs, specific for distinct subsets of *NPC* mRNAs, or other layers

of regulations also partake in the fine-tuning of the translation of these transcripts remains to be investigated.

While Hek2 represses *NPC* mRNA translation, protein turnover also contributes to the definition of the cellular levels of nucleoporins. Indeed, excess Nups likely synthesized in the absence of Hek2-dependent translational repression appear to be buffered by an increase in their degradation rates (Fig. 2f). This mechanism is reminiscent of the post-translational attenuation described to occur for multiprotein complex subunits when they are naturally produced in super-stoichiometric amounts[7], or overexpressed due to genomic amplification[6]. Excess subunits of NPCs, which do not assemble into stable complexes and could be possibly unfolded, are thereby expected to undergo increased ubiquitin-dependent, proteasome-mediated degradation. Several conserved ubiquitin ligases are susceptible to partake in this process, including (i) Hul5 and San1, which recognize misfolded proteins in the cytoplasm and the nucleus, respectively[41,42]; (ii) Tom1, which couples ubiquitin to unassembled ribosomal proteins[43]; or (iii) any yet-to-be characterized quality control factor specialized in the degradation of orphan polypeptides, as recently identified in mammals[44]. The fact that the cellular concentration of Hek2-regulated nucleoporins such as Nup59, Nup1 and Nup116 is tightly restricted by both translational repression and protein degradation suggests that their accumulation could be detrimental, with these hydrophobic proteins being potentially prone to form toxic aggregates. Consistently, we found that Nup1 can form cytoplasmic foci when Hek2 and proteasome functions are inhibited (Supplementary Fig. 2f), and overexpressed Nup59 was similarly reported to accumulate within cytoplasmic structures[45]. Interestingly, overproduction of Nup170, a direct partner of Nup59, was described to trigger the formation of cytoplasmic foci containing distinct unassembled NPC subunits[46], suggesting that these excess, mislocalized nucleoporins might also interfere with the NPC assembly process.

In agreement with the physiological importance of such Hek2-mediated regulations, it is not surprising that the activity of this protein is itself under control. We found that sumoylation of Hek2 occurs on two different domains, thus generating two distinct monosumoylated versions of the protein (Fig. 3b–d, Supplementary Fig. 3d, e). Both modified regions are located at the vicinity of the third K-homology (KH) domain (Supplementary Fig. 3d), the major RNA-interacting motif of the protein[24], providing a possible molecular rationale for the SUMO-mediated decrease in RNA binding scored in vivo (Fig. 3f, g, Supplementary Fig. 3g) and in vitro (Fig. 3i, j). In this respect, inhibition of RNA recognition could be caused by steric hindrance, as already reported for several sumoylated DNA- or RNA-binding proteins[32], or, alternatively, occurs through changes in the oligomerization status of the protein, as proposed in the case of human hnRNP C1[47]. Furthermore, the spatio-temporal control of Hek2 function is likely to depend on a combination of post-translational modifications including, besides its sumoylation, its reported phosphorylation by Yck1[29] and its ubiquitination detected in proteome-wide analyses[48]. Notably, Hek2 sumoylation appears to have significant effects at low stoichiometry, a paradox commonly observed for SUMO targets[12]. However, the real stoichiometry of Hek2 sumoylation may be under-estimated in view of the intrinsic difficulty to preserve this labile modification[49,50]. Alternatively, transient sumoylation may promote permanent changes in Hek2 association with RNA or yet-to-be identified protein partners that would be maintained after removal of the modification, as already shown for other factors[51]. Finally, the stoichiometry of sumoylation may be much greater for the small pool of Hek2 actually involved in RNA binding. In support of this last hypothesis, Hek2 recruitment onto mRNAs primarily occurs prior to nuclear export, as shown by its association with nuclear, partly unprocessed mRNPs (Fig. 3f, g)[27]; this Hek2

population, a minor fraction of this predominantly cytoplasmic protein (Supplementary Fig. 4a), would be the only one targeted by the nuclear sumoylation machinery[18]. Desumoylation by Ulp1 could then favor its binding onto mRNAs at the nucleoplasmic side of NPCs (Supplementary Fig. 4b). The cytoplasmic fate of certain mRNPs would then be determined prior to export, as in the case of *ASH1* whose asymmetrical localization and translation depends on Hek2 binding. This molecular mechanism could also explain why *ASH1* asymmetry requires Nup60[52], since this nucleoporin is one of the major determinants of Ulp1 stability at NPCs[15,16].

The control of Hek2 function through Ulp1-mediated desumoylation is also likely to adjust its RNA-binding activity in response to the status of nuclear pores in the cell. Since several distinct nucleoporin subcomplexes are indeed required to position and stabilize Ulp1 at the pore (Fig. 4a, b)[15,16], the level of activity of this SUMO protease provides a readout for the number and the functionality of NPCs. Consistently, changes in NPC composition in mutant or perturbed physiological situations impact Ulp1 activity and trigger the accumulation of sumoylated, inactive versions of Hek2 (Fig. 4e, f). In view of the function of Hek2 in controlling NPC mRNA translation (Fig. 2), this could in turn result in the increased synthesis of nucleoporins in a feedback process (Supplementary Fig. 4c). Their recruitment into NPCs would then compete their proteasomal degradation and contribute to restore NPC integrity. Strikingly, some of the nucleoporins that are targeted by this mechanism appear to be the most limiting ones for completing fully assembled NPCs (Supplementary Fig. 4d). Among them, Nsp1 is also critical to define NPC number during the asymmetric division of budding yeast[53,54]. While the pathway described here could indeed connect the cellular availability of specific nucleoporins to the status of NPCs, other quality control mechanisms are known to control NPC homeostasis. In yeast, aberrant NPC assembly intermediates are cleared from the nuclear envelope by the activity of ESCRT-III/Vps4 complexes[55], while in mammals, defects in the assembly of nuclear pore baskets triggers a cell cycle delay[56].

Localization of SUMO proteases at NPCs has been conserved in all eukaryotes[11] and also involves several distinct NPC-associated determinants in mammalian cells[57,58]. Sumoylation of KH domain containing Hek2 orthologs such as hnRNP K, hnRNP E1 and hnRNP E2 has also been reported[59–61]. Strikingly, hnRNP K desumoylation involves SENP2, the NPC-localized ortholog of Ulp1 in mammals[62]. In view of the association between hnRNP K and a subset of *NPC* mRNAs in a genome-wide survey of human RBPs[63], the conservation of the pathway described here will certainly deserve further investigation.

## Methods

**Yeast strains and plasmids.** Unless otherwise indicated, all the strains used in this study (listed in Supplementary Table 1) are isogenic to BY4742/BY4741 and were grown in standard culture conditions. Experiments using the *ulp1* allele were performed at semi-permissive temperature (30 °C) as previously described[35]. Experiments with the *ubc9* thermosensitive mutant were performed following 2 h of shift at 37 °C. When indicated, cycloheximide (0.1 mg per ml, Sigma), MG132 (100 μM, Sigma) or ethanol (10% v/v) were added to the medium for the indicated time. Drug-sensitive *erg6Δ* strains were used for MG132 treatment[16]. Construction of plasmids (listed in Supplementary Table 2) was performed using standard PCR-based molecular cloning techniques and was checked by sequencing.

**Bioinformatic analysis of RNA immunoprecipitation datasets.** RIP, CLIP or CRAC data were collected for the following RNA-binding proteins: Yra1 (RIP followed by microarray analysis, one replicate[22]), Nab2 (CRAC, three replicates[27]), Npl3 (RIP followed by microarray analysis, one replicate[23]), Nab4/Hrp1 (RIP followed by microarray analysis, one replicate[23]), Mex67 (CRAC, three replicates[27]), Sto1 (CRAC, three replicates[27]), Xrn1 (CRAC, two replicates[27]), Ski2 (CRAC, four replicates[27]), Mtr4 (CRAC, three replicates[27]) and Hek2 (CRAC, one replicate[27]; CLIP, one replicate[26]; RIP followed by microarray analysis, one replicate in two distinct studies[24,25]). For each dataset, all protein-coding RNAs were ranked and given a color according to their relative binding to the corresponding

RBP. Scores available from microarray or sequencing analyses[22–25] were used to split the RNAs in four equally sized groups corresponding respectively to "high" (light yellow), "medium" (dark yellow), "low" (dark blue) and "very low/no" (light blue) binding. CLIP data were used to define bound (light yellow) and unbound (light blue) mRNAs according to the published peak calling analysis[26]. CRAC hits were first normalized by hits per million within each RBP CRAC dataset, then for each mRNA ($\Sigma i^2 = 1$) to account for differences in mRNA abundances, and scaled to occupy the 0–1 range. Colors ranging from light blue (0) to light yellow (1) were used to depict the binding of a given mRNA to a RBP. Binding categories were further displayed for *NPC* mRNAs (Fig. 1a) or proteasome/exosome RNAs (Supplementary Fig. 1c). Gene set enrichment analyses were performed as previously described[64]. The MEME software (v4.11.3)[65] was applied to the sequences of *NUP59*, *NUP116*, *NUP1*, *NSP1* and *NUP100* mRNAs. Out of 6 retrieved motifs, 5 corresponded to FG-coding sequences, while one, found with an e-value of 4.8e−7, matched the known Hek2-binding site (Fig. 1d).

**mRNP and RNA immunoprecipitation.** Cbc2-pA- and Mlp2-pA-associated mRNPs complexes were purified as previously described[35]: cells were lysed by bead beating using a Fastprep (Qbiogene) in the following extraction buffer: 20 mM Hepes pH 7.5, 110 mM KOAc, 2 mM MgCl₂, 0.1% Tween-20, 0.5% Triton X-100, 1 mM dithiothreitol (DTT), 1× protease inhibitors cocktail, complete EDTA-free, Roche, and antifoam B, Sigma, 1:5000. After 10,000 × *g* centrifugation at 4 °C for 5 min, the soluble extract was incubated with IgG-conjugated magnetic beads for 10 min at 4 °C. Beads were washed 3 times with extraction buffer and eluted with sodium dodecyl sulfate (SDS) sample buffer.

Hek2-pA-associated mRNA purifications were performed according to the same procedure in the presence of RNAsin (Promega, 40 U per ml of buffer). Hpr1 RNA immunoprecipitation was performed as previously described[35]: cells were crosslinked with 1% formaldehyde for 10 min at 25 °C. Cells were further lysed by bead beating in the following lysis buffer: 50 mM Hepes pH 7.5, 140 mM NaCl, 1 mM EDTA, 1% Triton X-100, 0.1% deoxycholate, 1× protease inhibitors cocktail, complete EDTA-free, Roche. Soluble extracts were recovered following centrifugation at 10,000 × *g* for 5 min at 4 °C and immunoprecipitated overnight at 4 °C in the presence of anti-Hpr1 antibodies[35]. Immuno-complexes were captured on protein-G sepharose beads (GE Healthcare) and washed as follows: twice with lysis buffer, twice with lysis buffer containing 360 mM NaCl; twice with 10 mM Tris pH 8, 250 mM LiCl, 0.5% Nonidet-P40, 0.5% deoxycholate, 1 mM EDTA and once with 10 mM Tris-HCl pH 8, 1 mM EDTA. Elution was achieved through 20 min of incubation at 65 °C in the presence of 50 mM Tris pH 8, 10 mM EDTA, 1% SDS. The eluate was deproteinized with proteinase K (Sigma, 0.2 mg per ml) and uncrosslinked for 30 min at 65 °C. Total and immunoprecipitated RNAs were purified with the Nucleospin RNAII kit (Macherey Nagel) and reverse transcribed with Superscript II reverse transcriptase (Life Technologies). cDNAs were further quantified by real-time PCR with a LightCycler 480 system (Roche) according to the manufacturer's instructions. The sequences of the primers used for qPCR in this study are listed in Supplementary Table 3. Controls without reverse transcriptase allowed estimating the lack of contaminating DNA.

**Polysome profiling analysis.** The protocol was adapted from a published procedure[40]. A total of 100 ml cultures were grown in YPD media to midlog phase ($OD_{600} = 0.4$–$0.6$). Prior to harvest, cycloheximide (CHX) (Sigma) was added to final a concentration of 0.1 mg per ml. All subsequent procedures were carried out on ice with pre-chilled tubes and buffers. Cultures were cooled on ice and pelleted by centrifugation at 2600 × *g* for 5 min at 4 °C. Pellets were washed twice in 2.5 ml of ice-cold lysis buffer (20 mM Tris-HCl pH 8, 140 mM KCl, 1.5 mM MgCl₂, 1% (v/v) Triton X-100, 0.5 mM DTT, 0.1 mg per ml CHX and 1 mg per ml heparin), resuspended in 0.7 ml of ice-cold lysis buffer and lysed by bead beating using a Fastprep (Qbiogene, 3 × 30 s). Cell debris and glass beads were removed by centrifugation at 2600 × *g* for 5 min at 4 °C. The supernatant was transferred to a 1.5 ml tube and clarified by centrifugation at 10,000 × *g* for 10 min at 4 °C. 10 A₂₅₄ units of extract were layered onto an 11 ml 20–50% (wt/vol) sucrose gradient prepared in the lysis buffer without Triton X-100. The samples were ultra-centrifuged at 39,000 × *g* for 2.5 h at 4 °C in a SW41 rotor. The gradients were fractionated in 14 fractions of 0.9 ml using an ISCO fractionation system with concomitant measurement of A₂₅₄. Total lysates and fractions were supplemented with 50 μl of 3 M NH₄Ac, 5 ng of Luciferase RNA (Promega), 1 μl of Glycoblue (Ambion) and 1.2 ml of ethanol. Samples were vortexed and precipitated overnight at −20 °C. The pellets were collected by centrifugation at 10,000 × *g* for 10 min at 4 °C, washed once in 75% ethanol and resuspended in 100 μl DEPC-treated H₂O. RNAs were further purified using the Nucleospin RNAII kit (Macherey Nagel) following the RNA clean-up procedure. Equal volumes of all samples were reverse transcribed with Superscript II reverse transcriptase (Life Technologies) and cDNAs were further quantified by real-time PCR as described above.

**Recombinant protein production.** His and GST fusion proteins were expressed in Rosetta (DE3) *Escherichia coli* cells transformed with the corresponding plasmids and grown in LB medium supplemented with the required antibiotics. Expression of the recombinant proteins was achieved by submitting bacterial cultures to cold and chemical shocks (4 °C, 2% ethanol), and inducing them with 0.2 mM

isopropyl-β-D-thiogalactopyranoside at 23 °C for 4 h. Bacterial pellets were collected by centrifugation and frozen in liquid nitrogen. Pellets were resuspended either in His buffer (20 mM $Na_2HPO_4$ pH 7.5, 500 mM NaCl, 10 mM imidazole, 0.2% (v/v) Triton X-100, 1 mM $MgCl_2$, 1× protease inhibitors cocktail, Roche) or GST buffer (50 mM Tris-HCl pH 7.5, 0.1% (v/v) Triton X-100, 10 mM KCl, 10% glycerol, 1 mM DTT, 1× protease inhibitors cocktail, Roche), treated with 0.5 mg per ml lysozyme for 1 h at 4 °C and lysed by sonication. His-tagged proteins were further solubilized by adding 0.5% Sarkosyl for 15 min at 4 °C, followed by the addition of 0.8% Triton X-100. Lysates were cleared by centrifugation at 10,000 × $g$ for 20 min at 4 °C. His-tagged proteins were purified on Ni-NTA agarose (Qiagen) for 2 h at 4 °C. Beads were then washed twice with His buffer and eluted four times with the same buffer containing 500 mM imidazole and 1% Triton X-100. GST fusion proteins were purified in the presence of 550 mM NaCl on Gluthatione sepharose (GE Healthcare) for 1 h and 30 min at 4 °C. Beads were then washed three times with GST buffer containing 500 mM NaCl, and eluted four times for 15 min in 50 mM Tris-HCl pH 8, 500 mM NaCl, 0.1% (v/v) Triton X-100, 10% glycerol and 15 mM gluthatione. Following purification, His and GST fusion proteins were dialyzed overnight at 4 °C against 20 mM Hepes KOH pH 7.9, 0.1 M KCl, 0.1 mM DTT, and 10% glycerol was added before storage at −80 °C.

**In vitro RNA-binding assay**. In vitro RNA-binding assays were performed according to a published procedure[66]. Streptavidin dynabeads (Invitrogen) were washed three times in 0.1 M NaOH, 0.05 M NaCl and once in 0.1 M NaCl. Then, 2 μg of biotinylated RNA (encompassing Hek2-binding sites on *NSP1* (21–80) or *NUP116* (162–221) mRNAs or a sequence from *NUP133* (1429–1488); Integrated DNA Technologies) were bound to 10 μl of beads in RNA-binding buffer (5 mM Tris-HCl pH 7.5, 1 M NaCl, 40 U per ml RNAsin) for 30 min at room temperature. The conjugated beads were then washed four times in RNA-binding buffer and incubated in protein-binding buffer (50 mM Hepes pH 7.5, 100 mM NaCl, 1 mM $MgCl_2$, 10% glycerol, 0.5 mM DTT, 0.1 mM phenylmethylsulfonyl fluoride, 0.1% bovine serum albumin, 40 U per ml RNAsin) for 15 min at 4 °C for saturation. Beads were then incubated in protein-binding buffer containing 1 mg per ml heparin and ~2 pmol of recombinant Hek2 for 30 min at 4 °C. Beads were then washed five times with protein-binding buffer containing 1 mg per ml heparin and eluted in SDS sample buffer.

**Sumoylation assays**. SUMO conjugates were isolated from yeast cells expressing a His-tagged version of SUMO using nickel agarose denaturing chromatography as previously described[35]: 100 $OD_{600}$ of cells were lysed by bead beating in 6 M guanidine HCl, 100 mM sodium phosphate pH 8, 10 mM Tris-HCl, 0.1% Triton X-100, 10 mM beta-mercaptoethanol and 50 mM *N*-ethylmaleimide (Sigma). Clarified lysates were incubated with Ni-NTA agarose beads (Qiagen) for 2 h at room temperature. Beads were washed twice with lysis buffer and three times with 8 M urea, 100 mM sodium phosphate, 10 mM Tris-HCl pH 6.3 before proceeding to elution in 8 M urea, 200 mM Tris-HCl pH 6.8, 1 mM EDTA, 5% (w/V) SDS, 0.1% (w/v) bromophenol blue and 1.5% (w/V) DTT.

In vitro sumoylation was performed as previously reported[50]: briefly, 3 μg of recombinant Hek2 was mixed with 300 nM of recombinant E1 enzyme (Aos1/Uba2), 700 nM of recombinant E2 enzyme (Ubc9) and 10 mM of a mutated version of Smt3 (K11,15,19 R) less prone to form poly-SUMO chains, in the presence of 5 mM adenosine triphosphate in a sumoylation buffer (50 mM Bis-Tris pH 6.5, 100 mM NaCl, 10 mM $MgCl_2$ and 0.1 mM DTT). The reaction was then incubated for 3 h at 37 °C and either stopped by addition of SDS sample buffer or further used for in vitro RNA-binding assays.

**Protein extraction and western blot analysis**. Total protein extraction from yeast cells was performed by the NaOH–TCA lysis method[49]. Samples were separated on 10% or 4–12% SDS–polyacrylamide gel electrophoresis (SDS-PAGE) gels and transferred to nitrocellulose or polyvinylidene difluoride membranes. Western blot was performed using the following, previously validated antibodies: polyclonal anti-GLFG[67] (to detect Nup116), 1:500; polyclonal anti-FSFG[68] (to detect Nup1), 1:4000; polyclonal anti-Nup133[69], 1:500; monoclonal anti-Pab1 (clone 1G1, sc-57953, Santa-Cruz), 1:1000; polyclonal anti-SUMO[70], 1:2000; monoclonal anti-HA (clone 16B12, MMS-101P, Covance), 1:1000; monoclonal anti-GFP (clones 7.1 and 13.1, 11814460001, Roche Diagnostics), 1:500; monoclonal anti-GST (clone 4C10, MMS-112P, Covance), 1:1000; rabbit IgG-HRP polyclonal antibody (to detect protein-A-tagged proteins, Z0113, DakoCytomation), 1:5000. For Nup59-GFP and Ulp1-GFP detection, specificity of anti-GFP antibodies was confirmed using untagged strains. Quantification of signals was performed based on serial dilutions of reference samples using the ImageJ software.

**Gene expression analyses**. Total RNAs were extracted from yeast cultures using Nucleospin RNAII (Macherey Nagel). Reverse transcription and cDNA quantification were performed as described above for RNA immunoprecipitation. Transcriptome analysis was achieved using microarrays as previously reported[35]: the *hek2Δ* versus *wt* comparison was performed twice using independent samples and dye swap. The averaged log2 of the mutant/wild-type ratios and the standard deviation between the two replicates were calculated for each gene. The genes showing a standard deviation of >0.5 were removed from the dataset. Comparisons

of *hek2Δ* transcriptome with Hek2 and Nab2 binding profiles were realized using published datasets[23,24]. Transcripts were split in four equally sized groups corresponding respectively to "strong", "medium", "low" and "very low/no" binding. For each category, the log2 of the mutant/wild-type ratios of the different transcripts were represented as a box plot.

**Cell imaging**. The smFISH was carried out on fixed cells using Stellaris Custom Probe Sets and RNA FISH buffers, according to the manufacturer's instructions (Biosearch Technologies). For Hek2-GFP localization, cells were fixed with 0.1 M $KPO_4$ pH 6.4, paraformaldehyde 4% for 15 min and nuclei were stained with 4,6-diamidino-2-phenylindole (DAPI). Nup1-GFP and Ulp1-GFP localization was analyzed on live cells. Wide-field fluorescence images were acquired using a DM6000B Leica microscope with a 100×, NA 1.4 (HCX Plan-Apo) oil immersion objective and a CCD camera (CoolSNAP HQ; Photometrics). Z-stack sections of 0.2 μm were acquired using a piezo-electric motor (LVDT; Physik Instrument) mounted underneath the objective lens. Images were scaled equivalently and 3D-projected using ImageJ, and further processed with Photoshop CS6 13.0 ×64 software (Adobe). Nuclear envelope intensities were determined with ImageJ following subtraction of the cytoplasmic background.

**Statistics**. The experiments were not randomized and the investigators were not blinded to allocation during experiments and outcome assessment. No statistical methods were used to predetermine sample sizes; ($n$) values were chosen in accordance with standard practices in RNA analyses in yeast, correspond to the number of biological replicates (e.g., independent yeast cultures) and are indicated in the corresponding figure legends. Error bars correspond to standard deviations. The two-tailed Welch's t-test, which allows unequal variance, was used to compare RNA-binding efficiencies in vitro or in vivo (Figs. 1f and 3j; Supplementary Fig. 3g). The Mann–Whitney–Wilcoxon test was used to compare Ulp1 nuclear envelope intensities in different strains (Fig. 4b) and RNA expression fold changes upon *HEK2* deletion (Supplementary Fig. 2a, b). Standard conventions for symbols indicating statistical significance were used: *$P ≤ 0.05$; **$P ≤ 0.01$; ***$P ≤ 0.001$; N.S., not significant.

**Data availability**. The complete microarray data are available in the ArrayExpress database under accession number E-MTAB-6065 (https://www.ebi.ac.uk/arrayexpress/experiments/E-MTAB-6065/). The uncropped scans of the blot images shown in Figures are provided in the Supplementary Fig. 5. All the other data supporting the findings of this study are available within the paper and its supplementary information files, or from the corresponding author upon request.

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

## Acknowledgements

We thank P. Chartrand, C. Dargemont, V. Doye, V. Géli, M. Hochstrasser, E. Johnson, S. Wente and X. Zhao for reagents; H. Bretes, L. Fusée and G. Lelandais for help with strain construction, in vitro experiments and data normalization, respectively; D. Tollervey for CRAC analyses; A. Babour, V. Doye, T.H. Jensen and R. Rothstein for discussion and critical reading of the manuscript. This work was supported by: CNRS (to B.P.); Fondation ARC pour la Recherche sur le Cancer (to B.P.); Ligue Nationale contre le Cancer (to B.P.; fellowship to J.O.R.); Ecole Doctorale "Structure et Dynamique des Systèmes Vivants" (#576), Université Paris-Sud, Université Paris-Saclay (fellowship to J.O.R.); and the "Who am I?" laboratory of excellence (grant numbers ANR-11-LABX-0071, ANR-11-IDEX-0005-02, fellowship to J.O.R.).

## Author contributions

Conceptualization: J.O.R., B.P.; methodology: J.O.R., B.C., B.P.; investigation: J.O.R., M.B., B.C, F.D., B.P.; formal analysis: J.O.R., A.T., F.D., B.P.; writing: J.O.R., B.P.; visualization: J.O.R., B.P.; funding acquisition: B.P.; supervision: B.C., B.P.

## Additional information

**Competing interests:** The authors declare no competing interests.

