## [Peer Review File(PDF 137 kb) · Nature Communications]

Reviewers' Comments:

Reviewer #1:

Remarks to the Author:

This paper by Rouviere et al., investigates a potential mechanism that controls the synthesis of key members of the nuclear pore complex in response to transport dysfunction and provides a useful paradigm for investigating how the stoichiometry of multi-component protein subcomplexes are controlled. Most of the paper focuses on the functional consequences of an interesting molecular link that the authors uncover (by mining published data) between nup encoding mRNAs and the RNP component Hek2. Their data are supportive of a model in which Hek2 helps control the translation of these RNAs. Moreover, the authors identify sumoylation of Hek2 as a mediator of its interactions with the RNAs and further tie this sumoylation to defects in the NPC itself. This sets up an appealing model in which NPC dysfunction leads to an increase in nup production. Overall, this is a very interesting study that is carefully presented, well controlled and the data are of high quality. They show a remarkable experimental breadth that encompasses both in vivo and direct in vitro reconstitution (of sumoylation) strategies. While there are clearly many open questions, particularly with respect to the mechanisms of translation repression and the links between sumoylation of Hek2 and putative increases in nup production (which remain to be well established), I think that there is tremendous value to the conceptual advances made in this paper and I recommend its publication.

Reviewer #2:

Remarks to the Author:

Rouvière et al. report evidence that levels of a specific subset of NPC subunit (nucleoporin) mRNAs are regulated in the yeast *S. cerevisiae* by the hnRNP-like protein Hek2, a translational repressor. This was discovered through a careful analysis of public databases of RNA-RNA-binding protein interactions, most of which were validated here. The authors show that Hek2 binds directly to these specific nucleoporin mRNAs and appears to control their translational efficiency based on polysome distribution analyses. These nucleoporins are degraded more rapidly, however, counteracting their increased translation. The effects of Hek2 deletion on both translational derepression and increased degradation are quite small, so this 'fine-tuning' of nucleoporin expression by Hek2 might be of physiological importance only under certain as yet unknown conditions.

The authors also show that Hek2 is sumoylated and the extent of sumoylation depends on Ulp1, an NPC-associated SUMO protease. Evidence is presented that the SUMO-Hek2 isoforms are inhibited for target nucleoporin mRNA binding. Ulp1 reverses this, but NPC disruption reduces NPC-bound Ulp1 levels. These results lead to a model in which the NPC-associated Ulp1 senses NPC integrity which if reduced, allows more Hek2 to remain sumoylated; as a result, Hek2 binds less well to nucleoporin mRNAs, which become translationally derepressed and help restore normal NPC levels. Overall, I think the manuscript has some quite interesting data and an appealing model but might still be a little too preliminary in nature to be suitable for Nature Communications unless the points below can be addressed.

Specific questions and comments:

-Ribosome profiling might be worth considering to confirm and better quantify the sucrose gradient fractionation data.

-What is the fraction of Hek2 that is sumoylated? How would modification of 1% or less of Hek2 via this modification lead to sufficiently reduced RNA binding of NPC mRNAs by Hek2 in the cell to promote their translation? Can the SUMO-Hek2 form be detected without prior binding to a Ni++ matrix? These issues should be addressed.

-Related to the previous point: In Fig. 3A and elsewhere, a large amount of Hek2 sticks nonspecifically to the nickel column in the absence of His6-SUMO (in fact, a far larger amount of unmodified protein sticks compared to SUMO modified forms). Doubly tagged SUMO would allow two rounds of distinct affinity purifications and could remove the ambiguity in determining if the bands are truly SUMO-Hek2 species. At the very least, one should use a *ubc9* mutant to show the two slow-migrating species are both lost or reduced.

- I believe the *ulp1* mutant used is a *ts* allele. Were experiments with this mutant done at restrictive or (semi)permissive T? Also, I am not sure if the source of the *ulp1^{ts}* mutant is cited.

-There is a second yeast SUMO protease, Ulp2. Was a mutant of this gene tested for effects on Hek2 sumoylation?

-In Fig. 3I, J, there is no control using a nontarget RNA.

-In Fig. 4E, input levels not shown. I don't feel the nonspecifically bound Hek2 is a sufficient control for this.

Answers to Reviewers' Comments

Reviewer #1

This paper by Rouviere et al., investigates a potential mechanism that controls the synthesis of key members of the nuclear pore complex in response to transport dysfunction and provides a useful paradigm for investigating how the stoichiometry of multi-component protein subcomplexes are controlled. Most of the paper focuses on the functional consequences of an interesting molecular link that the authors uncover (by mining published data) between nup encoding mRNAs and the RNP component Hek2. Their data are supportive of a model in which Hek2 helps control the translation of these RNAs. Moreover, the authors identify sumoylation of Hek2 as a mediator of its interactions with the RNAs and further tie this sumoylation to defects in the NPC itself. This sets up an appealing model in which NPC dysfunction leads to an increase in nup production. Overall, this is a very interesting study that is carefully presented, well controlled and the data are of high quality. They show a remarkable experimental breadth that encompasses both in vivo and direct in vitro reconstitution (of sumoylation) strategies. While there are clearly many open questions, particularly with respect to the mechanisms of translation repression and the links between sumoylation of Hek2 and putative increases in nup production (which remain to be well established), I think that there is tremendous value to the conceptual advances made in this paper and I recommend its publication.

We acknowledge Reviewer #1 for their comments.

- Regarding the possible mechanisms of Hek2-mediated translation repression: an earlier report had demonstrated that Hek2 represses translation initiation of the *ASH1* mRNA by binding directly to the translation initiation factor eIF4G1, and likely constraining its initiation-promoting activity (Paquin et al, 2007). We now mention in the *Discussion* section (p11) the possibility that the same mechanism contributes to Hek2-dependent repression of *NPC* mRNAs translation.
- With respect to the links between Hek2 and Nup production: we now include an additional experiment showing that the combined inhibition of Hek2 and proteasome functions can enhance the formation of abnormal cytoplasmic foci of nucleoporins (**novel Supplementary Fig. 2F**). This supports the view that the fine-tuning of Nup production mediated by Hek2-dependent translational repression and protein degradation is important to prevent the formation of such potentially toxic aggregates.

Reviewer #2

Rouvière et al. report evidence that levels of a specific subset of NPC subunit (nucleoporin) mRNAs are regulated in the yeast *S. cerevisiae* by the hnRNP-like protein Hek2, a translational repressor. This was discovered through a careful analysis of public databases of RNA-RNA-binding protein interactions, most of which were validated here. The authors show that Hek2 binds directly to these specific nucleoporin mRNAs and appears to control their translational efficiency based on polysome distribution analyses. These nucleoporins are degraded more rapidly, however, counteracting their increased translation. The effects of Hek2 deletion on both translational derepression and increased degradation are quite small, so this 'fine-tuning' of nucleoporin expression by Hek2 might be of physiological importance only under certain as yet unknown conditions.

We acknowledge Reviewer #2 for this remark. We have now identified a physiological situation in which such a fine-tuning of nucleoporin production is likely relevant. Indeed, we now report that in conditions of disturbed proteostasis (proteasome inhibition), Hek2 is required to prevent the accumulation of cytoplasmic aggregates of Nup1, which is encoded by one of its target mRNAs (**novel Supplementary Fig. 2F**, see also answer to Reviewer #1).

The authors also show that Hek2 is sumoylated and the extent of sumoylation depends on Ulp1, an NPC-associated SUMO protease. Evidence is presented that the SUMO-Hek2 isoforms are inhibited for target nucleoporin mRNA binding. Ulp1 reverses this, but NPC disruption reduces NPC-bound Ulp1 levels. These results lead to a model in which the NPC-associated Ulp1 senses NPC integrity which if reduced, allows more Hek2 to remain sumoylated; as a result, Hek2 binds less well to nucleoporin mRNAs, which become translationally derepressed and help restore normal NPC levels. Overall, I think the manuscript has some quite interesting data and an appealing model but might still be a little too preliminary in nature to be suitable for Nature Communications unless the points below can be addressed.

Specific questions and comments:

- Ribosome profiling might be worth considering to confirm and better quantify the sucrose gradient fractionation data.

Polysomal profiling is the traditional "gold standard" to assess the degree of association of cellular mRNAs with ribosomes as a reliable quantification of translation from the mRNA point-of-view. Ribosome profiling rather determines the position of ribosomes at codon resolution, this positional information being critical to analyse translation from the ribosome perspective. However, this technique does not discriminate whether the short ribosome-protected fragments come from a highly-translated mRNA, or from several non-efficiently translated mRNA molecules (see *PMID: 25380596, 24893926*). While ribosome profiling could have provided novel insights into Hek2 function at the transcriptome level, this limitation prompted us to favor polysome fractionation in the scope of our study on *NPC* mRNAs.

-What is the fraction of Hek2 that is sumoylated? How would modification of 1% or less of Hek2 via this modification lead to sufficiently reduced RNA binding of NPC mRNAs by Hek2 in the cell to promote their translation? Can the SUMO-Hek2 form be detected without prior binding to a Ni⁺⁺ matrix? These issues should be addressed.

The observed SUMO-modified fraction of Hek2 is likely less than 5% of the total amount of the protein and is not detectable in whole-cell lysates without prior Ni⁺⁺ purification; however, these low steady-state levels of modification appear to be sufficient to reduce binding to NPC mRNAs *in vivo*, as proven by the ~2-fold decreased Hek2-RNA association scored in the *ulp1* mutant in distinct assays (**Fig. 3F-G** and **Supplemental Fig. 3G**). Such an apparent paradox is typical of the large majority of sumoylated proteins for which SUMO modification has significant effects at low stoichiometries, as pointed by several reviews in the field (see for instance *PMID: 15808504, 16338371, 23746258*).

Possible explanations are now detailed in the *Discussion* section (p12). First, the real stoichiometry of Hek2 sumoylation may be under-estimated in view of the intrinsic difficulty to preserve this labile modification during SUMO-conjugates isolation (*PMID: 19107412, 22562154*). Second, the stoichiometry of sumoylation may be much greater for the small pool of Hek2 actually involved in RNA binding. In support of this hypothesis, Hek2 recruitment onto mRNAs primarily occurs prior to nuclear export, as shown by its association with nuclear, partly unprocessed mRNPs (**Fig. 3F-G**; *PMID: 23993093*); this Hek2 population, a minor fraction of this predominantly cytoplasmic protein (**novel Supplemental Fig. 4A**), would be the only one targeted by the nuclear sumoylation machinery (*PMID: 23712011*). Finally, transient sumoylation may promote permanent changes in Hek2 association with RNA or yet-to-be identified protein partners that would be maintained after removal of the modification, as already shown for several other SUMO targets (reviewed in *PMID: 15189146, 19616654*). In the future, additional investigations of the dynamics of Hek2 sumoylation-desumoylation will be required to tackle these questions, but we believe that they are out of the scope of the present study.

-Related to the previous point: In Fig. 3A and elsewhere, a large amount of Hek2 sticks nonspecifically to the nickel column in the absence of His6-SUMO (in fact, a far larger amount of unmodified protein sticks compared to SUMO modified forms). Doubly tagged SUMO would allow two rounds of distinct affinity purifications and could remove the ambiguity in determining if the bands are truly SUMO-Hek2 species. At the very least, one should use a *ubc9* mutant to show the two slow-migrating species are both lost or reduced.

The non-specific binding of a fraction of non-sumoylated Hek2-HA is also observed in the absence of His-SUMO (see 2nd lanes in **Fig. 3B-C**) and is a classical issue in SUMO-conjugates purification (see several examples in *PMID: 16978391*), as mentioned in the legend to Fig. 3.

To further confirm that modified bands are indeed SUMO-Hek2 species, we performed the two following additional experiments:

- we purified in parallel SUMO-conjugates from cells expressing His-tagged SUMO or Flag-His-tagged-SUMO, and compared the migration of Hek2 modified species. Modified Hek2 species purified from cells expressing doubly tagged SUMO were migrating slightly slower than those from His-SUMO cells, and were not detected in the absence of tagged SUMO expression (**novel Supplementary Fig. 3C**).

- we analyzed Hek2 sumoylation in a *ubc9* thermosensitive mutant and found the modified bands to be lost upon *UBC9* inactivation (**novel Supplementary Fig. 3A**).

Together with the finding that modified bands are decreased/lost in *hek2-KR* mutants (**Supplementary Fig. 3E**) and specifically increased upon *ULP1* inactivation (see below), our data establish that modified Hek2 species are SUMO-conjugates.

- I believe the *ulp1* mutant used is a *ts* allele. Were experiments with this mutant done at restrictive or (semi)permissive T? Also, I am not sure if the source of the *ulp1ts* mutant is cited.

This information had indeed to be clarified in both the *Results* and the *Methods* sections: the *ulp1* mutant used in this study is the thermosensitive allele initially described by the Hochstrasser lab (*PMID: 10094048*, now referenced), and experiments were performed at semi-permissive temperature (30°C).

-There is a second yeast SUMO protease, Ulp2. Was a mutant of this gene tested for effects on Hek2 sumoylation?

We have now performed this experiment (**novel Supplementary Fig. 3B**). In contrast with *ULP1* loss-of-function, *ULP2* inactivation (*ulp2Δ* cells) does not trigger accumulation of sumoylated Hek2, but rather leads to lower levels of modified Hek2. This decreased sumoylation is likely caused by the reduced availability of conjugatable SUMO previously scored in *ulp2* mutant cells (*PMID: 10713161, 17724121*).

-In Fig. 3I, J, there is no control using a nontarget RNA.

We now provide additional control experiments for these *in vitro* RNA binding assays (**novel Fig. 1E-F** and **Fig. 3I-J**). Neither naïve beads (ϕ), nor beads coated with a biotinylated RNA probe encompassing a random sequence from the non-target *NUP133* mRNA pulled down recombinant Hek2 as efficiently as specific binding sequences did.

-In Fig. 4E, input levels not shown. I don't feel the nonspecifically bound Hek2 is a sufficient control for this.

Inputs are now presented in this figure (**Fig. 4E-F, bottom panels**).

Reviewers' Comments:

Reviewer #1:

Remarks to the Author:

The additional experimental evidence and discussion provided further strengthen what was already an excellent paper. I am very supportive of publication.

Reviewer #2:

Remarks to the Author:

This revision and the accompanying rebuttal letter do a nice job addressing my initial concerns. I am happy to support publication of the current version of the paper.